# Separation and Identification of Resveratrol Butyrate Ester Complexes and Their Bioactivity in HepG2 Cell Models

**DOI:** 10.3390/ijms222413539

**Published:** 2021-12-17

**Authors:** Ming-Kuei Shih, You-Lin Tain, Chiu-Min Cheng, Chien-Ning Hsu, Yu-Wei Chen, Hung-Tse Huang, Chi-I Chang, Chih-Yao Hou

**Affiliations:** 1Graduate Institute of Food Culture and Innovation, National Kaohsiung University of Hospitality and Tourism, Kaohsiung 812, Taiwan; mkshih@mail.nkuht.edu.tw; 2Department of Pediatrics, Kaohsiung Chang Gung Memorial Hospital, Chang Gung University College of Medicine, Kaohsiung 833, Taiwan; tainyl@hotmail.com; 3Institute for Translational Research in Biomedicine, Kaohsiung Chang Gung Memorial Hospital, Chang Gung University College of Medicine, Kaohsiung 833, Taiwan; 4Department of Aquaculture, National Kaohsiung University of Science and Technology, Kaohsiung 811, Taiwan; cmcheng@nkust.edu.tw; 5Department of Pharmacy, Kaohsiung Chang Gung Memorial Hospital, Kaohsiung 833, Taiwan; cnhsu@cgmh.org.tw; 6School of Pharmacy, Kaohsiung Medical University, Kaohsiung 807, Taiwan; 7Department of Medicine, Chang Gung University, Linkow 333, Taiwan; naosa720928@gmail.com; 8Department of Biochemical Science and Technology, National Taiwan University, Taipei 106, Taiwan; kk49310953@nricm.edu.tw; 9Department of Biological Science and Technology, National Pingtung University of Science and Technology, Pingtung 912, Taiwan; 10Department of Seafood Science, National Kaohsiung University of Science and Technology, Kaohsiung 811, Taiwan

**Keywords:** resveratrol butyrate ester (RBE) complex, separate, identified, antioxidant

## Abstract

Resveratrol butyrate ester (RBE) complexes have demonstrated higher antioxidant capacity and anti-fat accumulation activity in previous studies. In this study, silica gel, high-performance liquid chromatography, and 1H nuclear magnetic resonance were used for separation and identification of RBE complex components. With the exception of resveratrol, five different structures of ester derivatives were separated from silica gel: 3,4′-di-O-butanoylresveratrol (ED2, 18.8%), 3-O-butanoylresveratrol (ED4, 35.7%), 4′-O-butanoylresveratrol (ED5, 4.4%), 3,5,4′-tri-O-butanoylresveratrol (ED6, 1.5%), and 3,5-di-O-butanoylresveratrol (ED7, 0.7%). Among the ester derivatives obtained, ED2 and ED4 were the main ester derivatives in the RBE complex. Thus, the cellular antioxidant activities of the RBE mixture, ED2, and ED4 were evaluated. Results showed that the antioxidant capacity of ED2 and ED4 was higher than that of the RBE mixture, demonstrating that the number and position of butyrate esterification sites are related to cell survival rate and antioxidant capacity. This study is the first to report the successful isolation, structural identification, and cellular biological antioxidant activity of RBE complex derivatives, which are key characteristics for the potential practical application of RBE complexes.

## 1. Introduction

Resveratrol (RSV), a natural stilbenoid hydroxylated derivative of stilbene, is mainly present in red wine, grapes, rhubarb, and blueberries and confers multiple biological benefits, including the reduction of oxidative damage and inflammation, neuroprotection, chemoprevention, in addition to exerting anti-obesity, anti-aging, anticancer, and lipid/glucose metabolism-regulating effects [1,2,3,4,5,6,7]. However, the structural requirements for multiple biological properties of RSV have not been completely established, and conflicting results have been reported in the literature [8,9,10,11,12]. In addition, it is noteworthy that RSV has poor aqueous solubility and undergoes rapid phase II conjugative metabolism in the intestine and liver, which results in low bioavailability, which may subsequently limit its biological potency and application in vivo [13]. Various RSV-oriented analogs have been designed by introducing different substituents on the aromatic rings of RSV [14,15,16,17,18], and electron-donating groups were found to be more effective than other substituent groups. Selective functionalization of natural products allows for modifications that can improve biological activity, stability, availability, solubility, and metabolism [19,20,21].

Since the synthesis of resveratrol derivatives and their novel functional properties have been a research hotspot. On the other way, short-chain fatty acids (SCFA), especially indicating acetic (C2), propionic (C3), and butyric (C4) acids, have been proven to have benefits for human health that play an important role in the gut microbiota [22,23,24,25] and such as energy metabolism [26], the inflammatory response [27], adipose tissue formation [28], and liver metabolism [29]. In a previous study, we successfully enhanced the production and activity of the products of a Steglich esterification reaction [30] with N-ethyl-N’-(3-dimethylaminopropyl) carbodiimide (EDC) and 4-dimethyl aminopyridine (DMAP). We applied the improved Steglich esterification reaction and used short-chain fatty acids (SCFA), especially indicating acetic acid (C2), propionic acid (C3) and butyric acid (C4) to synthesize a brand new resveratrol with RSV Acetate (RAE), resveratrol propionate (RPE), and resveratrol butyrate (RBE) [31]. When the reaction was performed for the esterification of RSV and butyric acid to form resveratrol butyrate esters (RBEs), a 30% increase in the production of RBEs was observed [32]. In addition, to RBE formation, we reported that the esterification of RSV with butyrate produced RSV, RBE mono-ester, RBE di-ester, and RBE tri-ester, which also showed better H_2_O_2_-scavenging activity than RSV [31]. RBEs can effectively inhibit fatty acid-induced lipid accumulation in HepG2 cells, with effects similar to those of RSV being achieved at a lower dose [32]. RBEs protect against liver damage caused by BPA exposure during the peripregnancy period and the effect of the gut microbiota on the gut–liver axis in offspring [33]. RBEs suppress BPA-induced obesity in female offspring rats and exhibit excellent modulatory activity in the intestinal microbiota, with potential for application in perinatological research [34]. Although RBEs have greater antioxidant capacity than RSV [31], the ability to prevent fat accumulation in a liver cell culture model [32], and the ability to reduce the negative effects caused by BPA to offspring [33,34], there are some biologically active effects that have not been understood, including the observed reduced fat content in female offspring rats [34] and liver protection function in male offspring rats [33]. Thus, further separation, identification, and purification of compounds derived from the RBE complex is important for the future application of RBEs, with respect to biological activity, bioavailability, and hemodynamics.

Our series of studies synthesized resveratrol derivatives, established synthetic methods, and identified and evaluated their antioxidant properties and biological activities in vitro and in vivo experiments, but the information available regarding the extensive comparison of structural characterization in vivo has not been reported. In addition, there remains a crucial gap in understanding the relationship between the biological effects of each RBE monomer. Now we are further purifying and identifying individual structures, hoping to find a major derivative structure with potential for human health and provide a solution to the problem of low resveratrol bioavailability. Therefore, we use silica gel, high-performance liquid chromatography, and 1H nuclear magnetic resonance to separate and identify RBE complex components. To facilitate a wide range of applications and enhance bioactivity, we exposed HepG2 cells to each RBE monomer and/or H_2_O_2_ to investigate the effects of structural characterization on cytotoxicity, reactive oxygen species (ROS) generation, glutathione (GSH)/oxidized glutathione (GSSG), and superoxide dismutase (SOD) activity.

## 2. Results

### 2.1. Isolation and Identification

The reaction mixture (24 g) was chromatographed over a silica gel column (5 × 50 cm) and eluted with a CH_2_Cl_2_–acetone gradient of increasing polarity, followed by MeOH to obtain 126 fractions. According to the TLC results, the fractions were combined to form 20 major fractions (Fr. 1–17). Fr. 1 (210 mg) from the CH_2_Cl_2_–acetone (100:0) elution was further purified via high-performance liquid chromatography (HPLC) to obtain ED6 (3,5,4′-tri-O-butanoylresveratrol; 36 mg) using hexane/CH_2_Cl_2_ (1:1). Fr. 1 (122 mg) from CH_2_Cl_2_–acetone (99:1) elution was further purified via HPLC to obtain ED7 (3,5-di-O-butanoylresveratrol; 17 mg) using hexane/CH_2_Cl_2_ (1:1). Fr. 5 (0.8 g) from CH_2_Cl_2_–acetone (97:3) was further purified using a silica gel column (2 × 45 cm) and eluted with CH_2_Cl_2_–EtOAc (100:0–90:1) to obtain ED2 (3,4′-di-O-butanoylresveratrol; 450 mg). Fr. 11 (1.9 g) from CH_2_Cl_2_–acetone (9:1) elution was further purified using a silica gel column (3 × 45 cm) and eluted with CH_2_Cl_2_–EtOAc (80:1–4:1) to obtain 12 fractions (11A–11L). Fr. 11F was purified by crystallization using acetone to obtain ED4 (3-O-butanoylresveratrol; 856 mg). Fr. 13 (0.9 g) from CH_2_Cl_2_–acetone (4:1) elution was further purified using a silica gel column (3 × 45 cm) and eluted with CH_2_Cl_2_–EtOAc (60:1–2:1) to obtain nine fractions (13A–13I). Fr. 13H was purified by crystallization using acetone to yield ED5 (4′-O-butanoylresveratrol; 106 mg). Fr. 15 (0.8 g) from CH_2_Cl_2_–acetone (1:99) elution was further purified using a silica gel column (2 × 45 cm) and eluted with CH_2_Cl_2_–EtOAc (1:1–1:99) to obtain ED3 (320 mg).

For analyzing the physical characteristics of RSV-butyric acid products, EI-MS, IR, and nuclear magnetic resonance (NMR) spectra (both 1H NMR and 13C NMR) were evaluated. The relevant spectral data for the separation and identification of RBE mixtures are shown in Appendix A. The RBEs, including RBE mono-ester, RBE di-ester, and RBE tri-ester, were produced by RSV and butyric acid esterification. After separation and purification of the RBE mixture, the identified structure types were RBE mono-ester (ED4 and ED5), RBE di-ester (ED2 and ED7), and RBE tri-ester (ED6) (Figure 1).

#### 2.1.1. Resveratrol (Trans-3,5,4′-Trihydroxystilbene) (ED3) (**1**)

White amorphous powder; ^1^H-NMR (400 MHz, Acetone-*d*_6_): δ_H_ 8.63 (1H, s), 8.35 (2H, s), 7.40 (2H, d, *J* = 8.4 Hz, H-2′, 6′), 7.00 (1H, d, *J* = 16.4 Hz, H-8), 6.87 (1H, d, *J* = 16.4 Hz, H-7), 6.83 (2H, d, *J* = 8.4 Hz, H-3′, 5′), 6.53 (2H, d, *J* = 2.0 Hz, H-2, 6), 6.26 (1H, t, *J* = 2.4 Hz, H-4); ^13^C-NMR (100 MHz, Acetone-*d*_6_): δ 159.7 (s), 158.3 (s), 141.0 (s), 130.0 (s), 129.1 (d), 128.8 (d), 126.9 (d), 116.5 (d), 105.7 (d), 102.8 (d); EI-MS (70 eV) *m*/*z* (rel. int.): 228 ([M]^+^, 100), 211 (6), 199 (5), 181 (17), 157 (7). The relevant spectral data for the separation and identification of ED3 (**1**) are shown in Appendix A.

#### 2.1.2. 3-O-Butanoylresveratrol (ED4) (**2**)

White amorphous powder; ^1^H-NMR (400 MHz, CD_3_OD): δ_H_ 7.36 (2H, d, *J* = 8.8 Hz, H-2′, 6′), 7.01 (1H, d, *J* = 16.4 Hz, H-8), 6.85 (1H, d, *J* = 16.4 Hz, H-7), 6.80 (1H, t, *J* = 1.6 Hz, H-2), 6.76 (2H, d, *J* = 8.4 Hz, H-3′, 5′), 6.70 (1H, t, *J* = 1.6 Hz, H-6), 6.38 (1H, t, *J* = 1.6 Hz, H-4), 2.52 (2H, t, *J* = 7.2 Hz, 3-OCOC*H*_2_CH_2_CH_3_), 1.74 (2H, sest, *J* = 7.6 Hz, 3-OCOCH_2_CH_2_CH_3_), 1.03 (3H, t, *J* = 7.2 Hz, 3-OCOCH_2_CH_2_CH_3_); ^13^C-NMR (100 MHz, CD_3_OD): δ 173.9 (s), 159.5 (s), 158.6 (s), 153.3 (s), 141.6 (s), 130.5 (d), 130.1 (s), 129.0 (d), 125.9 (d), 116.5 (d), 111.6 (d), 111.3 (d), 108.6 (d), 36.9 (t), 19.4 (t), 13.9 (q); EI-MS (70 eV) *m*/*z* (rel. int.): 298 ([M]^+^, 43), 228 (100), 199 (4), 181 (11), 71 (9). The relevant spectral data for the separation and identification of ED4 (**2**) are shown in Appendix A.

#### 2.1.3. 4′-O-Butanoylresveratrol (ED5) (**3**)

White amorphous powder; ^1^H-NMR (400 MHz, CD_3_OD): δ_H_ 7.50 (2H, d, *J* = 8.8 Hz, H-2′, 6′), 7.03 (2H, d, *J* = 8.8 Hz, H-3′, 5′), 7.02 (1H, d, *J* = 16.4 Hz, H-8), 6.95 (1H, d, *J* = 16.4 Hz, H-7), 6.49 (2H, d, *J* = 2.4 Hz, H-2, 6), 6.21 (1H, t, *J* = 2.4 Hz, H-4), 2.52 (2H, t, *J* = 7.2 Hz, 3-OCOCH_2_CH_2_CH_3_), 1.73 (2H, sext, *J* = 7.2 Hz, 3-OCOCH_2_CH_2_CH_3_), 1.02 (3H, t, *J* = 7.6 Hz, 3-OCOCH_2_CH_2_CH_3_); ^13^C-NMR (125 MHz, Methanol-*d*_4_,): δ_C_ 173.8 (s), 159.7(s), 151.5 (s), 140.6 (s), 136.5 (s), 130.2 (d), 128.4 (d), 128.3 (d), 122.9 (d), 106.1 (d), 103.2 (d), 36.9 (t), 19.4 (t), 13.9 (q); EI-MS (70 eV) *m*/*z* (rel. int.): 298 ([M]^+^, 18), 229 (21), 228 (100), 181 (9). The relevant spectral data for the separation and identification of ED5 (**3**) are shown in Appendix A.

#### 2.1.4. 3,5-di-O-Butanoylresveratrol (ED7) (**4**)

White amorphous powder; ^1^H-NMR (400 MHz, CDCl_3_): δ_H_ 7.25 (2H, d, *J* = 8.4 Hz, H-2′, 6′), 7.02 (2H, d, *J* = 1.6 Hz, H-2, 6), 6.90 (1H, d, *J* = 16.0 Hz, H-8), 6.71 (1H, d, *J* = 16.0 Hz, H-7), 6.78 (2H, d, *J* = 8.4 Hz, H-3′, 5′), 6.74 (1H, t, *J* = 1.6 Hz, H-4), 2.55 (4H, t, *J* = 7.6 Hz, 3,5-OCOCH_2_CH_2_CH_3_), 1.78 (sest., 4H, *J* = 7.6 Hz, 3,5-OCOCH_2_CH_2_CH_3_),1.04 (t, 6H, *J* = 7.2 Hz, 3,5-OCOCH_2_CH_2_CH_3_); ^13^C-NMR (100 MHz, CDCl_3_): δ_C_ 172.4 (s), 156.2 (s), 151.1 (s), 140.2 (s), 130.3 (d), 128.8 (s), 128.0 (d), 124.1 (d), 116.6 (d), 115.6 (d), 113.6 (d), 36.1 (t), 18.3 (t), 13.6 (q); EI-MS (70 eV) *m*/*z* (rel. int.): 368 ([M]^+^, 31), 298 (30), 228 (100), 181 (7), 71 (12). The relevant spectral data for the separation and identification of ED7 (**4**) are shown in Appendix A.

#### 2.1.5. 3,4′-di-O-Butanoylresveratrol (ED2) (**5**)

White amorphous powder; ^1^H-NMR (400 MHz, CDCl_3_): δ_H_ 7.38 (2H, d, *J* = 8.4 Hz, H-2′, 6′), 6.88 (1H, d, *J* = 16.0 Hz, H-8), 6.79 (1H, d, *J* = 16.0 Hz, H-7), 7.02 (2H, d, *J* = 8.5 Hz, H-3′, 5′), 6.72 (1H, s, H-2), 6.64 (1H, s, H-6), 6.43 (1H, s, H-4), 2.52 (4H, t, *J* = 7.6 Hz, 3,4′-OCOCH_2_CH_2_CH_3_), 1.78 (sest., 4H, *J* = 7.2 Hz, 3,4′-OCOCH_2_CH_2_CH_3_),1.04 (t, 6H, *J* = 7.2 Hz, 3,4′-OCOCH_2_CH_2_CH_3_); ^13^C-NMR (100 MHz, CDCl_3_): δ_C_ 172.7 (s), 156.9 (s), 151.6 (s), 150.1 (s), 139.4 (s), 134.7 (s), 128.6 (d), 127.8 (d), 127.5 (d), 121.7 (d), 111.5 (d), 111.1 (d), 108.5 (d), 36.2 (t), 28.4 (t), 13.6 (q); EI-MS (70 eV) *m/z* (rel. int.): 368 ([M]^+^, 40), 298 (98), 228 (100), 199 (9), 181 (18), 71 (36). The relevant spectral data for the separation and identification of ED2 (**5**) are shown in Appendix A.

#### 2.1.6. 3,5,4′-tri-O-Butanoylresveratrol (ED6) (**6**)

White amorphous powder; ^1^H-NMR (400 MHz, CDCl_3_): δ_H_ 7.46 (2H, d, *J* = 8.8 Hz, H-2′, 6′), 7.09 (2H, d, *J* = 2.0 Hz, H-2, 6), 7.06 (2H, d, *J* = 8.8 Hz, H-3′, 5′), 7.04 (1H, d, *J* = 16.4 Hz, H-8), 6.95 (1H, d, *J* = 16.4 Hz, H-7), 6.79 (1H, t, *J* = 2.0 Hz, H-4), 2.53 (6H, t, *J* = 7.2 Hz, 3,5,4′ -OCOCH_2_CH_2_CH_3_), 1.77 (sest., 6H, *J* = 7.6 Hz, 3,5,4′-OCOCH_2_CH_2_CH_3_), 1.03 (t, 9H, *J* = 7.2 Hz, 3,5,4′ -OCOCH_2_CH_2_CH_3_); ^13^C-NMR (100 MHz, CDCl_3_): δ_C_ 172.2 (s), 171.7 (s), 151.3 (s), 150.4 (s), 139.4(s), 134.3 (s), 129.5 (d), 127.5 (d), 127.1 (d), 121.8 (d), 116.8 (d), 114.4 (d), 36.1 (t), 18.3 (t), 13.6 (q); EI-MS (70 eV) *m*/*z* (rel. int.): 438 ([M]^+^, 25), 368 (98), 298 (92), 228 (100), 199 (7), 181 (11), 71 (53). The relevant spectral data for the separation and identification of ED6 (**6**) are shown in Appendix A.

With the exception of resveratrol, five different structures of ester derivatives were separated from silica gel: 3,4′-di-O-butanoylresveratrol (ED2, 18.8%), 3-O-butanoylresveratrol (ED4, 35.7%), 4′-O-butanoylresveratrol (ED5, 4.4%), 3,5,4′-tri-O-butanoylresveratrol (ED6, 1.5%), and 3,5-di-O-butanoylresveratrol (ED7, 0.7%). The hydrophobic order of resveratrol butyrates is ED6 (tri-butyric acid derivatives) > ED2 and ED7 (di-butyric acid derivatives) > ED4 and ED5 (mono-butyric acid derivatives). Therefore, the solubility of ED6 in the organic solvent, dimethyl sulfoxide, is the best, followed by ED2 and ED7, and lastly by ED4 and ED5. Among the ester derivatives obtained, ED2 and ED4 were the main ester derivatives in the RBE complex. Thus, the cellular antioxidant activities of the RBE mixture, ED2, and ED4 were evaluated.

### 2.2. Antioxidant Properties of RBEs and Their Purified Monomer in HepG2 Cells

One of the most extensively investigated biological activities of RSV is its cancer chemopreventive potential, since Jang et al. demonstrated its ability to inhibit carcinogenesis at multistep stages [35]. Taking into account the causative role of free radicals (or reactive oxygen species) in inducing oxidative stress during cancer [36,37], we evaluated the antioxidant activity of RBEs and their purified monomers using assays that evaluated the following: intracellular ROS generation, intracellular GSH/glutathione disulfide (GSSG) ratio, and intracellular SOD activity. ED2 and ED4 are the main ester derivatives in the RBE complex from the separate percent obtained. Thus, the cellular antioxidant activity of RBEs, ED2, and ED4 was evaluated.

#### 2.2.1. Different Derivatives of RBEs Inhibited ROS Generation

The production and clearance of ROS in normal cells is in dynamic equilibrium. Appropriate ROS can promote immunophysiological functions [38]. However, the generation of excessive ROS causes cell damage [39]. Their scavenging capacity against ROS generation might explain the protective mechanism of antioxidants against free radical-induced cell apoptosis [40]. Thus, we determined the inhibitory effects of different derivatives of RBEs on ROS generation in H_2_O_2_-treated HepG2 cells. Figure 2 shows that HepG2 cells were treated with 0.2 mM H_2_O_2_ for 6 h (positive control; 100%). It can be seen that pre-treatment of HepG2 cells with 50 µM samples (RBEs, ED2, and ED4) for 24 h significantly reduced the intracellular ROS content to 28.2%, 51.6%, and 82.3%, respectively. This shows that ED2 and ED4 not only have antioxidant capacity, but that it is also significantly higher than that of RBEs. This result reveals two important points. First, monomers derived from different butyric acid quantities and positions have varying capacities to inhibit H_2_O_2_-induced ROS generation: single butyric acid-derived ED4 has a greater capacity than butyric acid-derived ED2. Second, the ROS generation inhibitory capacity of RBEs is significantly lower than that of ED2 and ED4. In addition, ED2 and ED4 are the two components with the highest content ratio. Therefore, it is speculated that other butyric acid-derived monomers of RBEs have different antioxidant properties, thus offsetting the effects of ED2 and ED4 in the inhibition of the generation of ROS induced by H_2_O_2_. Thus, among the five monomers of RBEs derived from mono-butyric acid, di-butyric acid, and tri-butyric acid, there were significant differences in the number and position of butyrate esterification sites, with respect to the biological activity of inhibiting the ability of H_2_O_2_ to induce production of ROS in HepG2 cells. The structure of mono-butyric acid derivative (ED4) was better suited for this activity than that of the di-butyric acid derivative (ED2).

#### 2.2.2. Different Derivatives of RBEs Decreased the Intracellular GSH/GSSG Ratio

GSH is a tripeptide with crucial roles in redox homeostasis, protection of proteins from irreversible oxidative modification, and detoxification of xenobiotics that are extruded from cells after conjugation with GSH by GSH-S-transferases [41]. The ratio between GSSG, the oxidized form of GSH, and total GSH levels has been frequently used as an index of the intracellular redox state. Figure 3A shows that HepG2 cells were treated with 50 µM samples for 24 h in advance, and RBEs significantly increased GSH content; however, ED4 treatment led to a significant decrease in GSH content (*p* < 0.05). H_2_O_2_ treatment in the GSH/GSSG ratio (Figure 3B) significantly increased GSSG content. At the same time, pretreatment of HepG2 cells with 50 µM samples (RBEs, ED2, and ED4) for 24 h significantly reduced the intracellular GSSG content, resulting in the GSH/GSSG ratio being significantly greater than that in the H_2_O_2_ group (*p* < 0.05), reaching 6, 5, and 28 times that of the H_2_O_2_ group, respectively. In this study, in the RBEs, ED2, and ED4 groups, the GSH content was not lower than that in the negative control group (Figure 3A). Therefore, compared with Figure 3B, it can be seen that pretreatment of RBEs, ED2, and ED4 can significantly protect HepG2 cells from damage by H_2_O_2_. The ability to reduce the oxidative damage of GSSG is significantly related to the number and position of butyrate esterification sites.

#### 2.2.3. Different Derivatives of RBEs Increased Intracellular SOD Activity

SOD activity is an important indicator of intracellular antioxidant power, which converts superoxide anions into H_2_O_2_ and O_2_ [42]. This study investigated SOD enzyme activity to determine the effects of different RBE derivatives on the intracellular antioxidant power of H_2_O_2_-treated HepG2 cells. As shown in Figure 4, after 6 h of exposure to 0.2 mM H_2_O_2_, HepG2 cells had significantly decreased SOD activity compared to the control group (*p* < 0.05). Pretreatment with different RBE derivatives significantly maintained intracellular SOD activity (*p* < 0.05). The effects of 50 µM RBEs, ED2, and ED4 on intracellular SOD activities were not significantly different compared to the negative control group (*p* > 0.05). Our results are similar to those of a previous study. Chen et al. (2021) found that SOD activity was significantly protected by pretreatment of HepG2 cells with GSH and *Porphyra haitanensis* hydrolysate-III (PHH-III) before H_2_O_2_ exposure [43]. Although the current research results indicate that RBEs, ED2, and ED4 may significantly maintain the SOD activity in HepG2 cells from H_2_O_2_ destruction, this ability was irrelevant with the number and location of butylated sites. However, this study cannot determine whether these compounds protect SOD activity because these compounds may eliminate ROS, so cells do not need to use SOD defense.

## 3. Discussion

Our previous studies have shown that modifying the chemical structure of RSV can affect its biological activity [31,32] and can significantly protect against BPA-induced disruption of liver function and intestinal bacteria and obesity in offspring rats [33,34]. Structural modifications of RSV have been made to the stilbene scaffold, such as the extension of the conjugated chain [44,45], incorporation of N-substituents of the carbon-carbon double bond [46], and the development of novel RSV analogs with cis-restricted conformation [47], which have resulted in significant enhancement of the antioxidant capacity. We previously completed the first report on the successful synthesis of RSV-butyric acid derivatives and elucidated the following: RBEs contain RSV derivatives of mono-butyrate, di-butyrate, and tri-butyrate [31,32], RBEs have better antioxidant capacity (in vitro test) [31] and inhibition of fat accumulation (cell mode) than RSV [32], and they have protective effects for the developmental origins of health and disease (DOHaD) in rat animal models (in vivo) [33,34]. Therefore, the overall focus of this study was to separate and purify RBEs and identify the mono-ester derivative components ED4 and ED5, the di-ester derivative components-ED2 and ED7, and the tri-ester derivative component ED6. ED2 and ED4 are the main ester derivatives in the RBE complex found from the separate percent obtained. Therefore, cellular antioxidant activity was evaluated to study the relationship between the number and position of butyrate esterification sites and antioxidant capacity among RBEs, ED2, and ED4.

Over the past few decades, considerable efforts have been devoted to studying the antioxidative activity of RSV. The phenolic OH groups, especially 4′-OH, and the trans conformation, are believed to be responsible for the excellent antioxidative activity of RSV [12,48,49,50,51]. The antioxidant properties of RSV are related to the hydroxyl group [52]. For example, the presence of 4′-OH and trans stereochemistry are involved in its inhibitory effect on cell proliferation [12].

In this study, the stilbene molecules were conjugated. Depending on the nature of the substituents, the phenol in stilbene may be saturated or absorbed by electrons. This may affect the electron donor/acceptor properties of stilbene derivatives, thereby affecting their antioxidant activity. Compared to ED4, esterification of 4′-OH from resveratrol produces ED2. Owing to this change, ED2 is more lipophilic than ED4. At a concentration of 50 μM, the inhibitory effect of ED2 on intracellular ROS production in HepG2 cells was not as effective as ED4 (Figure 2), but antioxidative activity was still associated with it. ED2 was a better antioxidant than RBEs (mixtures) at the same concentration of 50 μM (Figure 2). We speculate that the butylation of 4′-OH retains its antioxidant activity in the cell. Based on the results of this study, it is clear that the 4′-OH group is necessary for a powerful cellular antioxidant effect, which is consistent with the result of another study [53]. Stojanovič et al. (2001) reported such a finding for trans-RSV, indicating that its para-hydroxyl group dominated the radical scavenging efficiency, whereas its meta-hydroxyl groups showed only minor reactivity [54]. Owing to insufficient yield, although the ED6 and ED7 monomers were purified in this study, the para-hydroxyl group in the molecular structure was butylated. However, further research is needed to determine whether the inference about the antioxidant capacity of ED6 and ED7 is concordant with the inference made by Stojanovič et al. (2001) [54].

In addition, based on the determined fluorescence intensity (a.u.), their ability to inhibit ROS generation follows the order ED4 > ED2 > RBEs. In the mono-ester position, ED4 (with the retained 4′-OH) exhibited an electron-donating capability comparable to that of RSV [55], thus highlighting the importance of 4′-OH in the electron-donating capability of RSV. This result also supports the previous conclusion that 4′-OH is more active than the 3-OH and 5-OH in the antioxidant reaction of RSV [15,44,47]. In contrast, impairment of the conjugated links, such as hydrogenation of the aliphatic double bond, could significantly decrease its electron-donating ability. This is in agreement with a finding by Yang et al. that extension by conjugation for the stilbene scaffold of RSV is an efficient strategy to improve its antioxidant activity [56]. In this study, the position of the ED4 butyrate substitution is consistent with the structure reported by Mikulski et al. (2010) [57]. Mikulski et al. found that oligomers, glucosides of RSV, and trans-resveratrol-3-O-glucuronide exhibited better antioxidative activity than RSV [57]. Therefore, the results of this study, in which the inhibition of the generation of ROS in cells occurred, suggested the following: the monomers derived with different number and position of butyrate esterification sites have different ROS scavenging capabilities, thereby protecting cells treated with H_2_O_2_; ED4 derived from mono-butyric acid is better than the di-butyric acid derivative ED2; and other butyric acid-derived monomers contained in RBEs may offset the inhibitory ability of ED2 and ED4 on the generation of ROS induced by H_2_O_2_. In other words, among the five monomers of RBEs derived from mono-butyric acid, di-butyric acid, and tri-butyric acid, the number and position of butyrate esterification sites led to significant differences in the biological activity of inhibiting H_2_O_2_-induced ROS production in HepG2 cells. The butyric acid derivative structure (ED4) had a better ROS inhibitory activity than the di-butyric acid derivative structure (ED2).

The selenium-containing glutathione peroxidase (GPx) catalyzes the reduction of H_2_O_2_ [58]. This reaction requires reduced GSH as a cosubstrate, and GSH is oxidized to oxidized glutathione (i.e., GSSG). GSSG can also be reduced to GSH by glutathione reductase (GRD) by utilizing NADPH. Intracellular reduced GSH is a major antioxidant in the cellular defense against oxidative stress, and a decreased GSH/GSSG ratio generally indicative of oxidative stress. In this study, pre-treatment of HepG2 cells with 50 µM test samples for 24 h showed that RBEs significantly increased GSH content, while ED4 pre-treatment resulted in a significant decrease in GSH content (*p* < 0.05) (Figure 3A). However, comparing the GSH/GSSG ratios revealed that RBEs, ED2, and ED4 significantly reduced the intracellular GSSH content, resulting in a significantly higher GSH/GSSG ratio, compared to the H_2_O_2_ group (*p* < 0.05), with the ratio for the ED4 group being the highest (Figure 3B). As the test sample was pretreated for 24 h before reprocessing H_2_O_2_, we speculated that RBEs might possess the biological activity of enhancing GSH biosynthesis (Figure 3A), while ED2 and ED4 do not. In addition, ED4 significantly reduced the ability of GSH to be oxidized to GSSG (Figure 3B). As mentioned in the previous paragraph, ED4 (with the retained 4′-OH) exhibited an electron-donating capability comparable to RSV [55]. Therefore, we speculate that the reason why ED4 significantly reduced the ability of GSH to be oxidized to GSSG (Figure 3B) may be related to ED4 having a direct effect on scavenging the ROS, thereby reducing the ability of GSH to oxidize. GSH biosynthesis involves glutamate-cysteine ligase (GCL) and GSH synthetase [59]. Jain et al. demonstrated that H_2_S increased intracellular GSH production by upregulating the GCL catalytic subunit (GCLC) and GCL modifier subunit (GCLM) [60]. As RBEs are mixtures, not a single compound, this study cannot further explore the relationship between butylation and GSH biosynthesis. On the other hand, the pathway of cellular metabolism of H_2_O_2_ is related to GPx and GSH [58], while the reduction of oxidized GSSG is related to GRD and NADPH. As the ratio of ED4 in GSH/GSSG is much higher than that of ED2, we speculate that the antioxidant capacity of ED4 should be the same as that of the 4′-OH of the stilbene molecule, which has a better electronic conjugated structure [53,57].

SOD mainly metabolizes superoxide anions into H_2_O_2_; therefore, SOD activity is directly related to the antioxidant capacity of cells [42]. In this study, RBEs, ED2, and ED4 could effectively maintain the activity of total SOD (TSOD) by preventing H_2_O_2_-induced oxidative damage, and there was no significant difference with the negative control group (Figure 4). In other words, upon comparing the SOD activity of the negative control group, it was noted seen that pretreatment with RBEs, ED2, and ED4 did not increase SOD activity. This study cannot determine whether these compounds protect SOD activity because these compounds have the antioxidant capacity to eliminate ROS, so cells may not need to use SOD defense. Previous studies have reported that 25–100 µM RSV can promote the expression of SOD activity in HepG2 cells [61,62]. Hence, we speculate that the products of RSV and butyrate, ED2 and ED4, do not promote HepG2 cell SOD activity, but can significantly inhibit ROS production, thereby preventing the cytoplasmic and mitochondrial SOD from the oxidative damage caused by H_2_O_2_. Although the preliminary results of this study show that the mono-ester derivative component ED4 and di-ester derivative component ED2 cannot induce SOD activity, thus far, there have been no reports on the relationship between RSV molecular structure and SOD gene expression and enzyme activity. Therefore, this study aimed to identify different 3-OH, 5-OH, and 4′-OH position butyrate esterification structures, such as mono-ester derivative component ED5, di-ester derivative component ED7, and tri-ester derivative component ED6. RSV structural modification and SOD activity correlation are topics suitable for further investigation.

## 4. Materials and Methods

### 4.1. Chemicals and Reagents

Trans-RSV was purchased from TCI Development Co., Ltd. (Shanghai, China). n-Butyric acid was obtained from ACROS (Morris Plains, NJ, USA). N,N0-dicyclohexylcarbodiimide, EDAC, 4-dimethylaminopyridine, DMAP, H_2_O_2_, methylthiazolyldiphenyl-tetrazolium bromide (MTT); 3-(4,5-dimethyl-2-thiazolyl)-2,5-diphenyl-2H-tetrazolium bromide, 98%), isopropanol (99.9%), and dimethyl sulfoxide (DMSO, 99.9%) were purchased from Sigma-Aldrich (St. Louis, MO, USA). Tetrahydrofuran (THF) was supplied by Echo Chemical Co., Ltd. (Miaoli County, Taiwan). Dulbecco’s modified Eagle’s medium (DMEM) was supplied by Invitrogen Life Technologies (Carlsbad, MD, USA), fetal bovine serum was supplied by Gibco-BRL (New York, NY, USA), ROS assay kit was supplied by Abcam (#ab113851; Hercules, CA, USA), GSH/GSSG assay kit was supplied by Promega (V6611; Madison, WI, USA), and the CuZn/Mn Superoxide Dismutase (CuZn-SOD/Mn-SOD) Activity Assay Kit (Hydroxylamine Method) was supplied by Elabscience (E-BC-K022-M; Hercules, CA, USA).

### 4.2. RBE Mixture Synthesis, Isolation, and Identification

#### 4.2.1. Synthesis of RBEs

RBEs were synthesized according to the modified method of Neises and Steglich [30] and Tain et al. [32]. RSV was mixed with butyric acid in THF. Subsequently, EDC and DMAP were added, and the esterification reaction was performed for 48 h in the absence of light. Upon completion of the reaction, a substantial amount of distilled water was added, and the reaction mixture was filtered to obtain the precipitated RBEs, which were freeze-dried and stored at −20 °C. Novel RBEs were produced by the esterification of RSV and butyric acid and included RBE mono-esters, RBE di-esters, and RBE tri-esters.

#### 4.2.2. Physical Properties and Chemical Compositions of RSV and RSV-Butyric Esters

Techniques used for analyzing the physical characteristics of the RSV-butyric acid products, including FTIR and NMR (both 1H NMR and 13C NMR analyses), were performed using a Bruker AVANCE 600 MHz NMR spectrometer (Bruker, Billerica, MA, USA) with deuterated dimethyl sulfoxide (DMSO-d6) as the solvent at 30 °C. TGA analyses and HPLC quantitation of the butyric esters of RSV were similar to those reported in our previous study [32].

#### 4.2.3. Separation and Identification of RBE Complex

Optical rotation was measured using a JASCO DIP-180 digital spectropolarimeter. IR spectra were recorded using a Nicolet 510P FT-IR spectrometer. UV spectra were measured in MeOH using a Shimadzu UV-1601PC spectrophotometer. The NMR spectra were recorded in CDCl3 at room temperature on a Varian Mercury plus 400 NMR spectrometer with residual solvent resonance as an internal reference. The 2D NMR spectra were recorded using the standard pulse sequences. For EI-MS and HR-EI-MS, Finnigan TSQ-700 and JEOL SX-102A spectrometers were used, respectively. TLC was performed on silica gel 60 F254 plates (Merck, Darmstadt, Germany). Column chromatography was performed on silica gel (230 400 mesh ASTM, Merck). HPLC was performed using a Hitachi L-7000 chromatograph with a Bischoff RI detector (Leonberg, Germany). A normal phase column (LiChrosorb Si 60, 7 μm, 250 × 10 mm, Merck) was used for isolation.

### 4.3. Cell Culture and Oxidative Treatment

#### 4.3.1. Cell Culture

HepG2 cells (human hepatocellular carcinoma cells) were obtained from the American Type Culture Collection (ATCC; Manassas, VA, USA). The cells were maintained in DMEM with 10% fetal bovine serum, 100 U/mL penicillin, 100 μg/mL streptomycin (all from HyClone, Logan, UT, USA), 1% essential amino acids, and 1% GlutaMAX™ (both from Gibco, Grand Island, NY, USA) at 37 °C in a humidifying incubator containing 5% CO_2_. For subculture, HepG2 cells were harvested at 80–90% confluence. For cell treatment, HepG2 cells were grown to 90% confluency, and then the medium was replaced with serum-free medium containing various concentrations of RBEs (0–50 µM) for 24 h. Then, the medium was replaced with 0.2 mM H_2_O_2_, followed by incubation for 6 h. RBEs are pre-dissolved in DMSO to prepare a stock solution.

#### 4.3.2. Cellular Measurement of Oxidative Stress

To compare the ROS generated by 0.2 mM H_2_O_2_ and 50 µM RBEs treatments in biological environments, a dichlorofluorescein diacetate (DCFDA) assay was performed in HepG2 cells. RBEs were prepared at the tested concentrations in serum-free medium and added to the cells for 24 h, followed by substitution with medium containing 0.2 mM H_2_O_2_. After a minimum of 6 h of incubation, the medium was removed from wells of the 96-well plate. DCFDA was then added at a concentration of 25 μM and incubated in the wells for 45 min. Once DCFDA diffuses into the cells, it is deacetylated into a non-fluorescent compound [63]; this compound can then be oxidized by ROS, resulting in the production of dichlorofluorescein (DCF), which is highly fluorescent [63]. The DCFDA solution was removed and replaced with complete medium. Serial dilutions based on sample volumes were then introduced into each well and incubated for 12 h to yield DCF. Subsequently, the fluorescence of DCF was measured using the FLUOstar Omega microplate reader (Ortenberg, Germany) with 485 nm excitation and 520 nm emission wavelength settings, followed by analysis using Omega software.

#### 4.3.3. GSH/GSSG-Glo™ Assay

Estimation of total glutathione (GSH + GSSG) level, GSSG level, and GSH/GSSG ratio was carried out using the GSH/GSSG-Glo™ assay (Promega, Southampton, UK). RBEs were prepared at the tested concentrations in serum-free medium and added to the cells for 24 h. Then, the medium with 0.2 mM H_2_O_2_ was replaced. After a minimum of 6 h of incubation, the media were discarded and replaced with either total GSH lysis reagent or oxidized GSH lysis reagent, and the cells were agitated for 5 min. Both plates were kept at room temperature for 30 min before luciferin generation reagent was added. The plates were briefly agitated in an orbital shaker and allowed to equilibrate at room temperature for 15 min. Luminescence was measured using the FLUOstar Omega microplate reader (Ortenberg, Germany), and the GSH/GSSG ratio was calculated according to the manufacturer’s instructions.

#### 4.3.4. SOD Activity Assay

SOD activity was determined using a commercial RANSOD kit (Randox Laboratories). SOD plays a primary role in protection against ROS and catalyzes the dismutation of superoxide anion to H_2_O_2_ and molecular oxygen. This method employs xanthine and xanthine oxidase to generate superoxide radicals that react with 2-(4-iodophenyl)-3(4-nitrophenol)-5-phenyltetrazolium chloride to form a red formazan dye. The cells were seeded in 96-well plates at a density of 4 × 10^4^ cells/well and incubated at 37 °C in a 5% CO_2_ atmosphere. After 24 h of cultivation, RBEs were prepared at the tested concentrations in serum-free medium and added to the cells for 24 h. Then, the medium with 0.2 mM H_2_O_2_ was replaced. After a minimum of 6 h of incubation, the medium was removed, and the adherent cells were washed with ice-cold PBS. After dissociating the cells with trypsin and transferring them to Eppendorf tubes, centrifugation was performed at 7000*× g* for 10 min at 4 °C. Subsequently, the supernatant was aspirated, and 200 mL of RIPA lysis buffer with a protease inhibitor (1:1000) was added to each sample. The lysates were sonicated on ice for 10 s per sample and centrifuged again. The supernatants were quantified using BCA protein assay and SOD assay kits. Supernatants (avoiding the pellet) were used to determine SOD activity. Using a Shimadzu UV-1601PC (Duisburg, Germany) spectrophotometer, the absorbance was monitored continuously at 505 nm and 37 °C by the inhibition degree of the reaction. The results are expressed as U/mL of SOD.

### 4.4. Statistical Analyses

All analytical experiments were performed at least thrice, and three samples were analyzed for each test. Data were collected and analyzed using a one-way analysis of variance and Duncan’s test. Differences were considered significant at *p* < 0.05. All statistical analyses were performed using SPSS Statistics software (version 12.0, St. Armonk, NY, USA).

## 5. Conclusions

In this study, silica gel, HPLC, and H-NMR were used to successfully separate and identify five esterified derivatives of different structures: ED4 and ED5 (mono-butyric acid derivatives), ED2 and ED7 (di-butyric acid derivatives), and ED6 (tri-butyric acid derivatives). The analysis of cellular antioxidant activity confirmed that ED2 and ED4 have better antioxidant activity than RBEs, and in addition, their biological activity is related to whether 4′-OH is esterified by butyrate. Compared with RBEs and ED2, ED4 showed good intracellular ROS inhibition and GSH/GSSG redox capacity. In addition, ED4 protected the activity of intracellular SOD enzymes. This study not only used short-chain fatty acids to improve the bioavailability and biological activity of RSV, but it is also the first to report the isolation of butyric acid-derived RSV monomers, and it proves their potential for health care functions. In the future, we will further analyze the functions and roles of ED2 and ED4 in bioavailability (such as hemodynamics and intestinal absorption) and animal intestinal flora to evaluate whether RBEs have the potential for biomedical application.

## Figures and Tables

**Figure 1 ijms-22-13539-f001:**
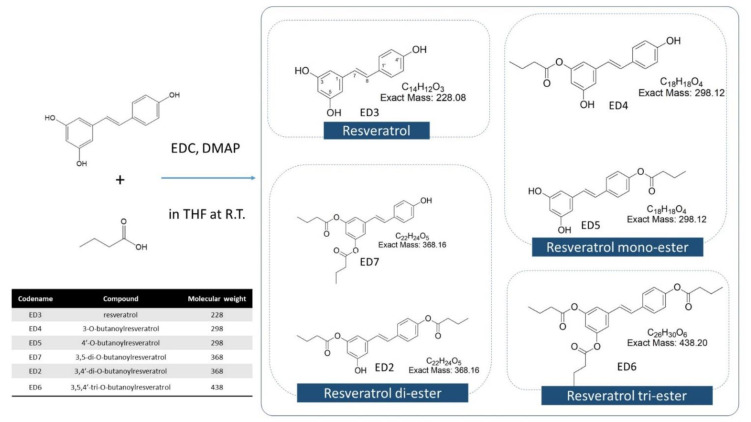
Structures in the resveratrol butyrate ester (RBE) mixture after separation and purification.

**Figure 2 ijms-22-13539-f002:**
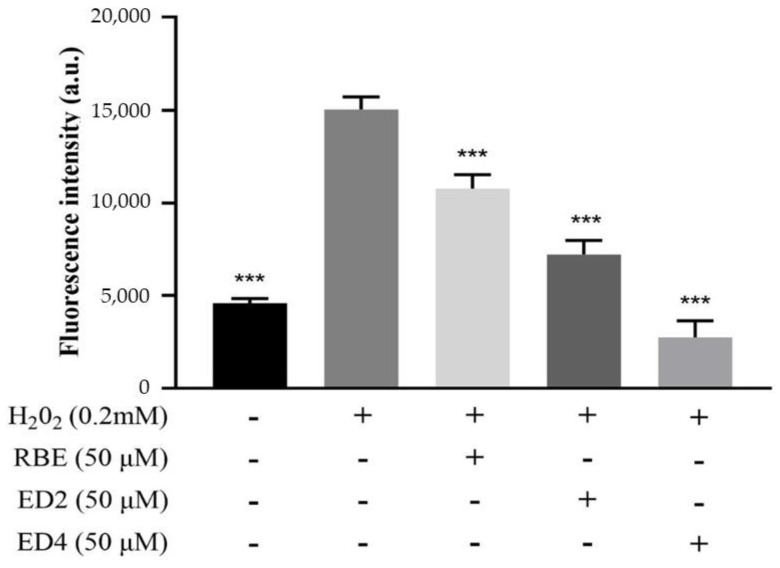
Effects of 50 µM of RBEs, ED2, and ED4 on reactive oxygen species (ROS) generation in H_2_O_2_-treated HepG2 cells. The data were obtained from 3–5 independent experiments and are presented as mean ± standard error of the mean (SEM). The annotation *** indicates a *p*-value < 0.001 versus H_2_O_2_-treated group.

**Figure 3 ijms-22-13539-f003:**
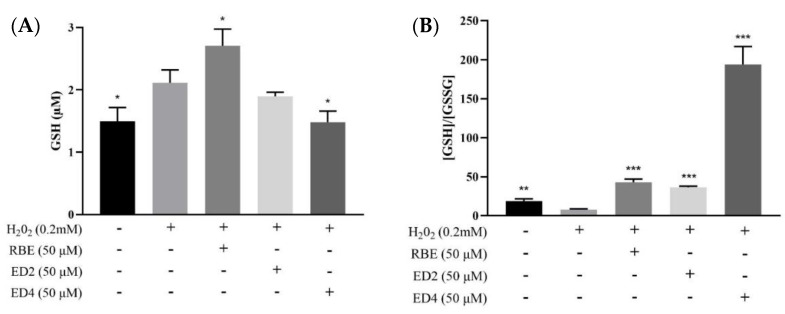
Effects of 50 µM of RBEs, ED2, and ED4 on the levels of GSH and GSH/GSSG, a potent endogenous antioxidative status in H_2_O_2_-treated HepG2 cells. (**A**) GSH levels; and (**B**) relative levels of GSH and oxidized glutathione (GSSG). Bars indicate the SEM. * *p* < 0.05; ** *p* < 0.01; *** *p* < 0.001 compared with the respective time control. Data were obtained from 3–5 independent experiments and are presented as mean ± SEM. The annotation *, **, *** indicates a *p*-value < 0.05, <0.01, <0.001 versus the H_2_O_2_-treated group, respectively.

**Figure 4 ijms-22-13539-f004:**
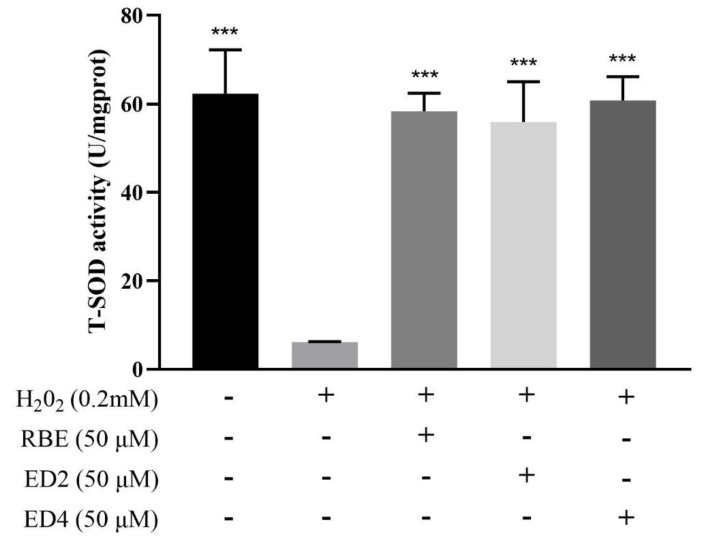
Effects of 50 µM of RBEs, ED2, and ED4 on SOD activity in H_2_O_2_-treated HepG2 cells. Data were obtained from 3–5 independent experiments and are presented as mean ± SEM. The annotation *** indicates a *p*-value < 0.001 versus H_2_O_2_-treated group.

## Data Availability

Not applicable.

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
