# Peer review of "Separation and Identification of Resveratrol Butyrate Ester Complexes and Their Bioactivity in HepG2 Cell Models"

_ijms, 2021, doi:10.3390/ijms222413539_

Round 1

Reviewer 1 Report

The paper ijms-1493247 deals with an interesting topic aimed at the isolation and characterization of antioxidant activities of RBE derivatives. The manuscript is well organized and can be interesting for readers. However, I have several comments that can inprove its quality.

Lines 57-77 in the Introduction bring the same information as the first paragraph of the Discussion. Please review these parts.

Legend to fig. 3 does not correspond to the data presented. The figure does not show the data concerning the SOD activity, does it?

The reference format throughout all manuscript should be unified, e.g. lines 72 and 306...etc.

line 433 - add supplier of the equipment (is it the same as in lines 446-447?)

line 435 - the sentence concerning MCF-7 cells should be deleted as in all study MCF-7 cells were not used

Author Response

Comments and Suggestions for Authors

The paper ijms-1493247 deals with an interesting topic aimed at the isolation and characterization of antioxidant activities of RBE derivatives. The manuscript is well organized and can be interesting for readers. However, I have several comments that can inprove its quality.

Thanks to the reviewer's constructive suggestion to improve the quality of this manuscript, the author would give the highest respect.

Lines 57-77 in the Introduction bring the same information as the first paragraph of the Discussion. Please review these parts.

Response:

Thanks for your suggestions.

The sentences have been rewritten.

Line 57-106.

Legend to fig. 3 does not correspond to the data presented. The figure does not show the data concerning the SOD activity, does it?

Response:

Thanks for your suggestions and apologies for the negligence.

The sentences have been rewritten.

Line 265-266.

The reference format throughout all manuscript should be unified, e.g. lines 72 and 306...etc.

line 433 - add supplier of the equipment (is it the same as in lines 446-447?)

Response:

Thanks for your suggestions.

The sentences have been rewritten.

Line 339; Line 467; Line478-479; Line497.

line 435 - the sentence concerning MCF-7 cells should be deleted as in all study MCF-7 cells were not used

Response:

Thanks for your suggestions and apologies for the negligence.

The sentences have been deleted.

Line 468.

Chih-Yao Hou, Dr.

Department of Seafood Science, National Kaohsiung University of Science and Technology, Kaohsiung 811, Taiwan

Reviewer 2 Report

The authors have synthesized butylated compounds of Resveratrol and isolated five monomers with biological activities from the mixture of synthetic products. They isolated five monomers with biological activities from the mixture of synthetic products, and focused on the activities of these monomers in enhancing antioxidant activity. The authors showed that two of the monomers contained in the mixture had more antioxidant activity than the mixture of resveratrol and its butylated compound, and revealed that those monomers will lead to future applications in therapeutic drugs and supplements.

 This paper is a quite interesting paper.

 However, there are a few points that need to be revised.

Specific comments;

1: Supplementary Materials could not be found online.

 Please provide the correct, URL. Alternatively, the description of Figure S1〜Figure S24(P3, line 109〜110), Figure S1〜Figure S4(P3, line 121), Figure S5〜Figure S8(P3, line 131) , Figure S9〜Figure S12(P3, line 141), Figure S13〜Figure S16(P4, line 151), Figure S17〜Figure S20(P4, line 161) and Figure S21〜Figure S24(P4, line 171)in the text should be deleted or the relevant results should be added to the text.

2: There are two abbreviations for oxidized glutathione in the text, GSSG and GSSH.

 Please unify to one of them; GSSG is more preferable.

 If author need to use GSSH, then you need to explain its definition in the text.

 There are many GSSH notations, including the following;

 P2, line 85,

 P6, line 222, line 224

 P9, line 334

 P10, line 339

3: The number of the Resveratrol products in Figure 1 does not match the number of the compound name in the manuscript.

 Please correct it.

 ・Resveratrol mono-esters are supposed to be ED4 and ED5, but in Figure 1 they are 2 and 3.

 Resveratrol is supposed to be ED3, but in Figure 1 it is 1.

Resveratrol de-esters are supposed to be ED7 and ED2, but in Figure 1 they are 4 and 5.

4: Please write A and B in the figures of Figure 3.

Also, please provide Figure legend correctly. There are three description in P7, line 234((A) Glutathione disulfide levels; (B) total glutathione levels; ( C ) ratio between glutathione disulfide and total glutathione ), but only two figures. Please delete the unnecessary description or provide the correct description.

5: P7, line 250: Describe the official name of PHH-III.

6: P8, line 259: Describe the official name of BPA.

7: P11, line 433: The manufacturer of the FLUOstar Omega microplate reader needs to be listed.

8: P11, line 436: MCF-7 cells are not described in 4.3.1. Please describe the nature and type of cells, or if not, delete this word.

9: P15, line 586: Correct the typing in Reference 35. H<inf>2</inf>O…….

Author Response

The authors have synthesized butylated compounds of Resveratrol and isolated five monomers with biological activities from the mixture of synthetic products. They isolated five monomers with biological activities from the mixture of synthetic products, and focused on the activities of these monomers in enhancing antioxidant activity. The authors showed that two of the monomers contained in the mixture had more antioxidant activity than the mixture of resveratrol and its butylated compound, and revealed that those monomers will lead to future applications in therapeutic drugs and supplements.

This paper is a quite interesting paper.

However, there are a few points that need to be revised.

Thanks to the reviewer's constructive suggestion to improve the quality of this manuscript, the author would give the highest respect.

Specific comments;

1: Supplementary Materials could not be found online.

Please provide the correct, URL. Alternatively, the description of Figure S1〜Figure S24(P3, line 109〜110), Figure S1〜Figure S4(P3, line 121), Figure S5〜Figure S8(P3, line 131) , Figure S9〜Figure S12(P3, line 141), Figure S13〜Figure S16(P4, line 151), Figure S17〜Figure S20(P4, line 161) and Figure S21〜Figure S24(P4, line 171)in the text should be deleted or the relevant results should be added to the text.

Response:

Thanks for your suggestions.

The Figure S1〜Figure S24 have been re-uploaded.

2: There are two abbreviations for oxidized glutathione in the text, GSSG and GSSH.

Please unify to one of them; GSSG is more preferable.

If author need to use GSSH, then you need to explain its definition in the text.

There are many GSSH notations, including the following;

P2, line 85,

P6, line 222, line 224

P9, line 334

P10, line 339

Response:

Thanks for your suggestions and apologies for the negligence.

The sentences have been deleted.

Line 109; Line 254; Line 256; Line 367; Line 372.

3: The number of the Resveratrol products in Figure 1 does not match the number of the compound name in the manuscript.

Please correct it.

・Resveratrol mono-esters are supposed to be ED4 and ED5, but in Figure 1 they are 2 and 3.

Resveratrol is supposed to be ED3, but in Figure 1 it is 1.

Resveratrol de-esters are supposed to be ED7 and ED2, but in Figure 1 they are 4 and 5.

Response:

Thanks for your suggestions.

The Figure 1 have been redrawn.

4: Please write A and B in the figures of Figure 3.

Also, please provide Figure legend correctly. There are three description in P7, line 234((A) Glutathione disulfide levels; (B) total glutathione levels; ( C ) ratio between glutathione disulfide and total glutathione ), but only two figures. Please delete the unnecessary description or provide the correct description.

Response:

Thanks for your suggestions and apologies for the negligence.

The Figure 3 have been redrawn.

The sentences have been rewritten.

Line 265-267.

5: P7, line 250: Describe the official name of PHH-III.

Response:

Thanks for your suggestions.

Corrected.

Line 283.

6: P8, line 259: Describe the official name of BPA.

Response:

Thanks for your suggestions.

Corrected.

Line 79.

7: P11, line 433: The manufacturer of the FLUOstar Omega microplate reader needs to be listed.

Response:

Thanks for your suggestions.

The sentences have been rewritten.

Line 467; Line478-479; Line497.

8: P11, line 436: MCF-7 cells are not described in 4.3.1. Please describe the nature and type of cells, or if not, delete this word.

Response:

Thanks for your suggestions and apologies for the negligence.

The sentences have been deleted.

Line 468.

9: P15, line 586: Correct the typing in Reference 35. H<inf>2</inf>O…….

Response:

Thanks for your suggestions.

Corrected.

Line 635-636.

Chih-Yao Hou, Dr.

Department of Seafood Science, National Kaohsiung University of Science and Technology, Kaohsiung 811, Taiwan

Reviewer 3 Report

In my opinion, the results presented in this paper should be the second part of the presented results. The study lacks information on how resveratrol esters were obtained. The presented evaluation of the isolation and identification of resveratrol esters is an incomplete research study. It is difficult for me to understand what is the final message of the publication because the presented  isolation and methods for determining the antioxidant potential of resvertrol esters do not constitute a complete research cycle. For this reason, I recommend the authors to present the entire sequence of research, from the synthesis of compounds, to testing their identity, ending with the assessment of biological activity. In this form, the work, in my opinion, is not suitable for publication.

Author Response

Comments and Suggestions for Authors

In my opinion, the results presented in this paper should be the second part of the presented results. The study lacks information on how resveratrol esters were obtained. The presented evaluation of the isolation and identification of resveratrol esters is an incomplete research study. It is difficult for me to understand what is the final message of the publication because the presented  isolation and methods for determining the antioxidant potential of resvertrol esters do not constitute a complete research cycle. For this reason, I recommend the authors to present the entire sequence of research, from the synthesis of compounds, to testing their identity, ending with the assessment of biological activity. In this form, the work, in my opinion, is not suitable for publication.

Thanks to the reviewer's constructive suggestion to improve the quality of this manuscript, the author would give the highest respect.

Response:

Thanks for your suggestions.

We corrected the content to further demonstrate the entire research sequence, especially from synthesizing compounds to testing their properties and evaluating the biological activity. The sentences have been rewritten. Line 57-106; Line 199-205

Chih-Yao Hou, Dr.

Department of Seafood Science, National Kaohsiung University of Science and Technology, Kaohsiung 811, Taiwan

Reviewer 4 Report

In this paper, Shih et al present new molecules issued from resveratrol and test their antioxidant abilities.

Overall, this paper is well built but there are several aspects which need to be reworked:

  • In figure 1, it would be nice to tag each molecule with its name (which is ED4 ? ED2 etc) for the non-chemist reader
  • A very important limit of resveratrol is its solubility and hence bioavailability. A comparison of solubility between the tested compounds would strengthen the interest for this work. Moreover, we don’t know which vehicul was used to solubilize the compounds for cell culture. Solubility can also be predicted based on chemical structure
  • Figs 2 and 3: it would be ideal to compare the antioxidant potentials of the molecules to resveratrol itself.
  • Several wording are not appropriate:
    • “Pretreatment with different RBE derivatives significantly protected against intracellular SOD activity” Rather, the molecules maintained SOD activity…
    • “The result in the current study shows that RBEs, ED2, and ED4 can significantly protect SOD from H2O2 destruction in HepG2 cells, and this ability is not related to the number and position of butyrate esterification sites.” I wasn’t aware that H2O2 destroyed SOD, besides the authors quantify SODD activity, not integrity.
    • “abilities to inhibit the ROS generated by H2O2” there is no proof of ROS generation inhibition. Rather, the compound scavenge the ROS, hence protecting the cells.
    • “ED4 significantly reduced the ability of GSH to be oxidized to GSSH” Does ED4 have a direct effect on GSH, or maybe it scavenged the ROS before GSH was needed ?
    • “In this study, RBEs, ED2, and ED4 350 could effectively protect the activity of total SOD (TSOD) by preventing H2O2-induced oxidative damage, and there was no significant difference with the negative control group (Figure 4).”

Not sure that the compounds protected the activity, rather that they scavenged the ROS, hence the cell didn’t have to use SOD…

Minor comments:

-Line 250, PHH-II is not explained

- Line 267 “RBEs have better antioxidant capacity (in vitro test) [24] and fat accumulation ability (cell mode) than RSV” actually ref 24 shows that RBEs have less fat accumulation

- Line 289: “the inhibitory effect of ED2 on intracellular ROS production in HepG2 cells was worse than that of ED4” ‘worse’ is not the best word here. Simply say it’s not as effective perhaps ?

- “which is consistent with the result another study [45]” rather ‘of another study’

- Line 334: “the intracellular GSSH content, resulting in a significantly higher GSH/GSSG ratio” careful: GSH, nor GSSH

Author Response

In this paper, Shih et al present new molecules issued from resveratrol and test their antioxidant abilities.

Thanks to the reviewer's constructive suggestion to improve the quality of this manuscript, the author would give the highest respect.

Overall, this paper is well built but there are several aspects which need to be reworked:

In figure 1, it would be nice to tag each molecule with its name (which is ED4 ? ED2 etc) for the non-chemist reader

Response:

Thanks for your suggestions.

The Figure 1 have been redrawn.

A very important limit of resveratrol is its solubility and hence bioavailability. A comparison of solubility between the tested compounds would strengthen the interest for this work. Moreover, we don’t know which vehicul was used to solubilize the compounds for cell culture. Solubility can also be predicted based on chemical structure

Response:

Thanks for your suggestions.

The sentences have been rewritten.

Line 461-463.

Figs 2 and 3: it would be ideal to compare the antioxidant potentials of the molecules to resveratrol itself.

Response:

Thanks for your suggestions. Since BREs (ED2, ED4, ED5, ED6, ED7) are the first compounds synthesized and isolated, their antioxidant potential can be better clarified if RSV can be used as a reference. We will compare the antioxidant potential of RBEs and RSV in future studies.

Several wording are not appropriate:

“Pretreatment with different RBE derivatives significantly protected against intracellular SOD activity” Rather, the molecules maintained SOD activity…

Response:

Thanks for your suggestions.

The sentences have been rewritten.

Line 277-278.

“The result in the current study shows that RBEs, ED2, and ED4 can significantly protect SOD from H2O2 destruction in HepG2 cells, and this ability is not related to the number and position of butyrate esterification sites.” I wasn’t aware that H2O2 destroyed SOD, besides the authors quantify SODD activity, not integrity.

Response:

Thanks for your suggestions.

The sentences have been rewritten.

Line 282-287.

“abilities to inhibit the ROS generated by H2O2” there is no proof of ROS generation inhibition. Rather, the compound scavenge the ROS, hence protecting the cells.

Response:

Thanks for your suggestions.

The sentences have been rewritten.

Line 351.

“ED4 significantly reduced the ability of GSH to be oxidized to GSSH” Does ED4 have a direct effect on GSH, or maybe it scavenged the ROS before GSH was needed ?

Response:

Thanks for your suggestions.

The sentences have been rewritten.

Line 374-378.

“In this study, RBEs, ED2, and ED4 350 could effectively protect the activity of total SOD (TSOD) by preventing H2O2-induced oxidative damage, and there was no significant difference with the negative control group (Figure 4).”

Not sure that the compounds protected the activity, rather that they scavenged the ROS, hence the cell didn’t have to use SOD…

Response:

Thanks for your suggestions.

The sentences have been rewritten.

Line 394-396; Line 397-400.

Minor comments:

-Line 250, PHH-II is not explained

Response:

Thanks for your suggestions.

Corrected.

Line 282.

Line 267 “RBEs have better antioxidant capacity (in vitro test) [24] and fat accumulation ability (cell mode) than RSV” actually ref 24 shows that RBEs have less fat accumulation

Response:

Thanks for your suggestions and apologies for the negligence.

The sentences have been rewritten.

Line 303.

Line 289: “the inhibitory effect of ED2 on intracellular ROS production in HepG2 cells was worse than that of ED4” ‘worse’ is not the best word here. Simply say it’s not as effective perhaps ?

Response:

Thanks for your suggestions and apologies for the negligence.

The sentences have been rewritten.

Line 323-324.

“which is consistent with the result another study [45]” rather ‘of another study’

Response:

Thanks for your suggestions and apologies for the negligence.

The sentences have been rewritten.

Line 328.

Line 334: “the intracellular GSSH content, resulting in a significantly higher GSH/GSSG ratio” careful: GSH, nor GSSH

Response:

Thanks for your suggestions and apologies for the negligence.

The sentences have been rewritten.

Line 109; Line 254; Line 256; Line 369; Line 374.

Chih-Yao Hou, Dr.

Department of Seafood Science, National Kaohsiung University of Science and Technology, Kaohsiung 811, Taiwan

Round 2

Reviewer 3 Report

The work requires general improvement

Author Response

Comments and Suggestions for Authors

The work requires general improvement

Thanks again to the reviewer for the great guidance and insightful suggestions to improve the quality of this article, and the author would give the highest tribute.

You and three other committee members have reviewed this manuscript. Therefore, in addition to following your insights, this manuscript has also been revised with careful reference to the opinions of other members. As a result of this, respectfully explain this situation to you.

The research works of the manuscript have been rewritten and improved.

Line 62-86; Line192-196.

Reviewer 4 Report

The authors extensively modified the manuscript and significantly improved it.

However I am confunded by the introduction, which is now very long and highly detailed. This is likely from another reviewer's comment, however summarizing may be needed

The authors have not brought additionnal information regarding the difference in solubility of their compounds.

Author Response

Comments and Suggestions for Authors

The authors extensively modified the manuscript and significantly improved it.

However I am confunded by the introduction, which is now very long and highly detailed. This is likely from another reviewer's comment, however summarizing may be needed

The authors have not brought additionnal information regarding the difference in solubility of their compounds.

Thanks again to the reviewer for the great guidance and insightful suggestions to improve the quality of this article, and the author would give the highest tribute.

The research works of the manuscript have been rewritten and improved.

Line 62-86; Line192-196.

Round 3

Reviewer 3 Report

The authors introduced information about the synthesis of compounds into the manuscript, the research of which is presented later in the work.
The reviewer recommends the work for publication as it does not constitute a coherent whole, so it may be worthwhile for the Authors to change the title of the work, suggesting that it is a continuation of the research.

Author Response

Comments and Suggestions for Authors

The authors introduced information about the synthesis of compounds into the manuscript, the research of which is presented later in the work.
The reviewer recommends the work for publication as it does not constitute a coherent whole, so it may be worthwhile for the Authors to change the title of the work, suggesting that it is a continuation of the research.

Thanks again to the reviewer for the great guidance and insightful suggestions to improve the quality of this article, and the author would give the highest tribute.

The title was changed to “Separation and identification of resveratrol butyrate ester complexes and their bioactivity in HepG2 cell models: from novel synthesis resveratrol ester derivative”

Reviewer 4 Report

the authors have answered my comments

Author Response

Comments and Suggestions for Authors

the authors have answered my comments

Thanks again to the reviewer for the great guidance and insightful suggestions to improve the quality of this article, and the author would give the highest tribute.
